# Cardiotoxicity of Chemotherapy: A Multi-OMIC Perspective

**DOI:** 10.3390/jox15010009

**Published:** 2025-01-08

**Authors:** Yan Ma, Mandy O. J. Grootaert, Raj N. Sewduth

**Affiliations:** 1VIB-KU Leuven Center for Cancer Biology, VIB, 3000 Leuven, Belgium; yan.ma@kuleuven.be; 2Faculty of Medicine and Dentistry, UC Louvain, Avenue Hippocrate 55, 1200 Woluwe-Saint-Lambert, Belgium; mandy.grootaert@uclouvain.be; 3Department of Cardiovascular Sciences, Centre for Molecular and Vascular Biology, KU Leuven, Herestraat 49, 3000 Leuven, Belgium

**Keywords:** cardiotoxicity, chemotherapy, cardiovascular diseases, cancer

## Abstract

Chemotherapy-induced cardiotoxicity is a critical issue in cardio-oncology, as cancer treatments often lead to severe cardiovascular complications. Approximately 10% of cancer patients succumb to cardiovascular problems, with lung cancer patients frequently experiencing arrhythmias, cardiac failure, tamponade, and cardiac metastasis. The cardiotoxic effects of anti-cancer treatments manifest at both cellular and tissue levels, causing deformation of cardiomyocytes, leading to contractility issues and fibrosis. Repeated irradiation and chemotherapy increase the risk of valvular, pericardial, or myocardial diseases. Multi-OMICs analyses reveal that targeting specific pathways as well as specific protein modifications, such as ubiquitination and phosphorylation, could offer potential therapeutic alternatives to current treatments, including Angiotensin converting enzymes (ACE) inhibitors and beta-blockers that mitigate symptoms but do not prevent cardiomyocyte death, highlighting the need for more effective therapies to manage cardiovascular defects in cancer survivors. This review explores the xenobiotic nature of chemotherapy agents and their impact on cardiovascular health, aiming to identify novel biomarkers and therapeutic targets to mitigate cardiotoxicity.

## 1. Introduction

### 1.1. Chemotherapy-Induced Cardiac Dysfunction

Cardio-oncology is an emerging field, as clinicians now have access to a large cohort of cancer survivors. While most cancer patients die of cancer itself, death from cardiovascular complications is the second cause of death in cancer patients, due to toxicity of both chemotherapy or radiotherapy. This is well described in the case of lung cancer [1], as recent publications indicate that cardiac dysfunction occurs secondarily in patients, at short or long term after the onset of cancer [2,3,4] (Table 1A). Arrhythmias, cardiac failure [5], tamponade [6], as well as cardiac metastasis [7,8] are common occurrences in lung cancer patients. However, the causality between these cardiovascular abnormalities and lung cancer is still unclear.

Alterations from anti-cancer treatments can manifest at the cellular level, with cell damage leading to cardiomyocyte dysfunction, but also occur at the tissue level, resulting in reduced contractility and fibrosis. Fibrosis can lead to stiffening of the heart muscle, impairing its ability to contract or relax properly. Recent studies show that repeated irradiations associated with chemotherapy in patients can cause secondary cardiac disease. Patients with radiotherapy have a higher risk of valvular, pericardial or myocardial disease [9]. Chemotherapy drugs such as the cytostatic agent cisplatin leads to coronary events (Table 1B). Additionaly, Tyrosine kinase inhibitors including EGFR inhibitors can include cardiomyopathy and heart failure, arrhythmias, coronary artery disease, pericardial effusions, and valvular heart disease [10]. The immune checkpoint inhibitors, pembrolizumab or nivolumab, cause cardiac failure and hypertension. The kinase inhibitors targeting BRAF (dabrafenib) or MEK (trametinib) are associated with hypertension and coronary disease [11]. Inhibitors targeting tumor angiogenesis through VEGF signaling can also cause cardiovascular defects, such as severe bleeding [12]. Proteasome inhibitors, such as bortezomib and carfilzomib, are essential in treating multiple myeloma but have been linked to cardiovascular dysfunction. These inhibitors can cause hypertension, heart failure, and arrhythmias by disrupting protein homeostasis in cardiomyocytes and vascular smooth muscle cells [13]. This adds to the complexity of the handling of cardiovascular diseases in patients with cancer.

**Table 1 jox-15-00009-t001:** Cardiovascular risk in cancer patients and patients having received specific chemotherapies. (A) Increased lethality in patients due to cardiovascular (CV) risk factors with different types of cancer, a follow-up to 120 months after the cancer diagnosis, cohort of approximately 140,000 patients, data adapted from [14] (B) Cardiac abnormalities caused by chemotherapy such as cytostatic agents, tyrosine kinases inhibitors (including EGFR inhibitors), VEGF and MEK/BRAF inhibitors, immune-checkpoint inhibitors and proteasome inhibitors adapted from [10].

**A Cardiovascular risk factors In cancer survivors**				
Dyslipedemia		32.1%		
Hypertension		26.4%		
Cardiomyopathy		20.8%		
Diabetes		13.2%		
Atherosclerosis		5.7%		
**B**	**Drugs**	**Cardiac dysfunction**	**Coronary events**	**Hypertension**
**Cytostatic agents**	Cisplatin, Gemcitabine, Taxanes	+	+++	+
**EGFR inhibitors**	Erlotinib		+++	
	Gefitinib		+	
	Afatinib	+		
	Trastuzumab	+++		
**Other Tyrosine kinase inhibitors**	Dasatinib	+++		
	Sunitinib	+		+++
	Sorafenib	+	+++	
**VEGF inhibitors**	Bevacizumab	+	+++	+++
	Nintedanib		+++	+++
**Immune checkpoint inhibitors**	Pembrolizumab, Nivo-lumab, Durvalumab, Atezolizumab	+++	+++	
**MEK/BRAF inhibitors**	Trametinib	+		+++
	Dafrafenib	+		+++
**Proteasome inhibitors**	Bortezomib Carfilzomib	++		++

+ stands for occasional, ++ for common and +++ for very common.

On top of the drugs listed here, a widely-used historical class of anti-cancer drugs, anthracyclines, such as doxorubicin (DOX) and daunorubicin, also cause major heart dysfunction. These drugs intercalate DNA strands and inhibit nucleic acid synthesis in cancer cells, but also can cause acute, subacute, or chronic cardiotoxicity, including arrhythmias, myocarditis, and heart failure [15]. Antimetabolites, such as 5-fluorouracil (5-FU), a pyrimidine analog, inhibits thymidylate synthase and depletes thymidine, leading to DNA synthesis inhibition. It can cause cardiotoxicity through endothelial dysfunction, increased homocysteine levels, and pro-inflammatory pathways [16].

### 1.2. Monitoring of Cardiotoxicity Strategies

Patients undergoing chemotherapy and radiotherapy face numerous challenges, including cardiac side effects that can significantly impact their quality of life. Ensuring patients receive adequate nutrition is crucial for maintaining their strength and immune function, and dietitians can provide personalized meal plans to address specific needs and side effects. Using techniques like dose fractionation and hyperfractionation can minimize damage to healthy tissues while effectively targeting cancer cells. Implementing advanced radiation techniques, such as Intensity-Modulated Radiation Therapy, can precisely target tumors and spare surrounding healthy tissue. Frequent assessments of blood counts, organ function, and overall health are necessary to detect and manage complications early [17]. While these preventive and interventional measures, healthcare providers can significantly improve the overall well-being and treatment outcomes for patients undergoing chemotherapy and radiotherapy, monitoring of cardiotoxicity is still needed.

Several hospitals have now developed different guidelines for the handling of cancer patients having received radiotherapy or specific chemotherapies (Figure 1A,B). In the case of chest/breast radiotherapy, the patients present a higher risk of atherosclerotic disease, pericarditis, tamponade or valvular disease. In the case of chemotherapy with Tyrosine kinases, MEK or BRAF inhibitors (Trastuzumab, Sunitinib, Sorafenib), a decrease of left ventricular ejection fraction causing heart failure, requires different treatments, ranging from specific drugs to surgery in some extreme cases [18]. So far, the therapeutic options currently available for the handling of cardiac abnormalities in cancer patients are limited, such as Angiotensin-converting enzyme (ACE) inhibitors, beta-blockers (BB), and the hydralazine-nitrate combination, which mitigate the symptoms and signs, with little or no demonstrated impact on cardiomyocyte death [19]. This highlights the need for better and more targeted therapies to reduce cardiovascular defects in cancer survivors, highlighting how multi-OMICs analysis of the heart tissue after chemotherapy could inprove our understanding of cardiotoxicity (Figure 1C).

### 1.3. Known Mechanism of Action of Cardiotoxicity

Different mechanisms of action and cardiotoxic effects associated with various anticancer drugs have been proposed (Table 2) [1]. Anthracyclines disrupt DNA replication and increase reactive oxygen species (ROS) generation, leading to mitochondrial dysfunction and cardiomyocyte apoptosis. Antimetabolites, such as 5-FU induce oxidative stress and inflammation, altering biochemical pathways and causing myocytic degeneration and endothelial injury. Alkylating agents, including cyclophosphamide and cisplatin, modify DNA, RNA, and proteins, resulting in severe oxidative stress and inflammation, contributing to cardiotoxicity [16]. Microtubule inhibitors, like paclitaxel, inhibit mitosis by altering microtubule polymerization, leading to endothelial damage and cardiotoxicity [10]. Additionally, tyrosine kinase inhibitors (TKI), such as imatinib, interfere with signal transduction and angiogenesis, causing cardiotoxic effects, particularly in treatments for HER2-positive breast cancer [20]. Lastly, Proteasome inhibitors, such as bortezomib and carfilzomib, lead to protein aggregation, endoplasmic reticulum stress, and impaired autophagy, which collectively contribute to cardiotoxicity and can alter nitric oxide signaling, further impacting cardiac function [21,22].

Dexrazoxane is a chelating agent used primarily as a cardioprotective drug during chemotherapy with anthracyclines [23]. It works by binding to iron and preventing the formation of free radicals that cause oxidative damage to the heart. This helps reduce the severity and incidence of cardiotoxicity associated with anthracycline treatment. However, there are some concerns about its use. Some studies suggest that dexrazoxane might reduce the anticancer efficacy of anthracyclines. Additionally, there is a potential risk of secondary malignancies, particularly in pediatric patients. This indicates that further research is needed into the pathways that are dysregulated upon cardiotoxicity. Multi-OMICs can provide such an unbiased and quantitative analysis, thereby facilitating the search for therapeutic targets to mitigate chemotherapy-induced cardiotoxicity.

## 2. Main Text

### 2.1. OMICs Analyses for Unbiased Analysis of Cardiotoxicity

OMICs analyses, encompassing genomics, transcriptomics, proteomics, and metabolomics, provide a comprehensive and unbiased approach to studying cardiotoxicity [24]. Genomic analyses involve examining DNA sequences to identify genetic variations that may predispose individuals to cardiotoxicity. By studying single nucleotide polymorphisms (SNPs) and other genetic markers, researchers can pinpoint genetic risk factors and potential therapeutic targets. Integrating data from genomics, transcriptomics, proteomics, and metabolomics offers a holistic view of the molecular mechanisms underlying cardiotoxicity. This integrated approach enhances our understanding and helps in developing strategies to mitigate the adverse effects of cardiotoxic agents. Here, we collected multi-OMICs datasets from databases such as Gene Expression Omnibus (GEO) from NIH. Each dataset are refered to using their GEO set expression (GSE) number, and significantly differentially expressed genes. The different analyses were done using ENRICHR from Maayan laboratory [25] by inputing the gene names, as well as fold changes for each significantly diffentially expressed gene when comparing control and drug-treated condition of the indicated cell type.

#### 2.1.1. Transcriptomics Analyses of the Anthracyclines Signature in Cardiomyocytes

Cardiomyocytes are vital for the heart’s contractile function. Anthracyclines, such as doxorubicin, cause cardiotoxicity manifesting as left ventricular systolic dysfunction, due to cardiomyocyte damage and fibrosis. Additionally, cardiac remodeling promotes tumor growth, as transverse aortic constriction (TAC) favors primary tumor growth and cardiac dysfunction promotes metastasis through activating transcription factor 3 (ATF3) [26]. Consistently to the reports of cardiotoxicity in patients treated with anthracyclines, transcriptomics analyses indicate activation of pro-inflammatory pathways such as the, JUN and NFKB1/RELAand EGR1 pathways. (Figure 2A). Also the transcription factors SP1 and SP3 were found enriched which are known to enhance or repress the activity of promoters of genes involved in differentiation, cell cycle progression, and oncogenesis [27]. Moreover, TP53, a key player in the DNA damage response, was found enriched as well as the growth regulatory and tumor repressor gene HIC1.

A recent CRISPR screen indicates alteration in single guide (sg) RNAs targeting DHX38, a gene related to DNA damage repair, or SMARCB1, a subunit of the SWI/SNF chromatin remodeling complex related to NGF signaling, as well as HOXB1, a homeobox DNA-binding domain related to TGF-β pathway, in cardiomyocytes after treatment with anthracyclines (Figure 2B). Pathway analysis of the inhibited sgRNAs showed changes in the NGF pathway, and pro-inflammatory pathways, such as STAT3 and neutrophin regulated ones (Figure 2C,D). These findings underscore the cardiotoxic effects of anthracyclines on cardiomyocytes through inducing DNA damage and inflammation, highlighting the alteration in critical signaling pathways like NFKB1, SP1/3 and TGF-β signaling.

In parralel to these unbiased OMICs findings, recent papers showed dependency to FOXO2, GATA6 [28], as well as NRF2 and AMPK signaling [29] for cardioprotection upon doxorubicin treatment. FOXO2 and GATA6 are transcription factors that play crucial roles in regulating gene expression related to cell survival and stress responses. Their involvement suggests that modulating these factors could enhance the heart’s resilience to the toxic effects of doxorubicin. Furthermore, Nebivolol is a cardioselective beta-blocker with antioxidant, anti-apoptotic, and vasodilator properties, commonly used to treat hypertension and heart failure. Recent research has highlighted its potential in preventing anthracycline-induced cardiotoxicity [30].

#### 2.1.2. Transcriptomics Analysis of the Anthracyclines Cardiotoxicity Signatures in Cardiac Endothelial Cells (ECs)

Endothelial cells are involved in maintaining vascular health and supporting the overall function of the heart by regulating vascular tone and permeability, angiogenesis, monocyte/leucocyte adhesion and platelet aggregation [31]. Anthracyclines can induce endothelial dysfunction by generating ROS, thereby impairing vasodilation and promoting inflammation. Additionally, anthracyclines activate endothelial cells, causing the exposure of phosphatidylserine on their surface and increasing the activity of tissue factor and other pro-coagulation factors, favoring thrombosis [32]. These effects contribute to the overall cardiovascular toxicity associated with anthracycline chemotherapy, highlighting the need for strategies to protect endothelial cells during cancer treatment [33].

RNA sequencing from human cardiac ECs after exposition to doxorubicin (GSE226116) revealed similar SP3, NFκB, RELA and SP1 signatures (Figure 3A), also indicating a proinflammatory response, similar to the one observed on cardiomyocytes. Proangiogenic signatures, such as a Robo4/VEGF one, and profibrotic ones were also observed (Figure 3B). These findings emphasize the dual impact of anthracyclines on both endothelial cells and cardiomyocytes, highlighting the need for targeted therapies that mitigate cardiovascular toxicity while maintaining anti-cancer efficacy.

#### 2.1.3. Transcriptomics Analysis of the Sorafenib/Sunitinib Cardiotoxicity Signatures

Sorafenib, is a kinase inhibitor used to treat advanced kidney cancer, liver cancer, and thyroid cancer. It works by blocking the action of certain proteins that promote cancer cell growth. Sorafenib targets multiple kinases involved in tumor cell proliferation and angiogenesis, which helps to slow the growth and spread of cancer cells.

However, sorafenib can cause several side effects, including cardiac issues. Patients may experience heart problems such as chest pain, fast heartbeats, trouble breathing, and swelling around the midsection or lower legs. It can also lead to hypertension, thrombosis, and other forms of cardiac toxicity. The current hypothesis about oxidative stress and inflammation is that TKI treatment blocks HER2 signaling which increases ROS and activates pro-apoptotic pathways [34]. Wang et al. found that afatinib, sorafenib, and ponatinib raised ROS levels and cardiac troponin T2 in cardiac cells [35]. Similarly, Bouitbir et al. showed that sunitinib reduced mitochondrial membrane potential, decreased GSH, increased H_2_O_2_, and activated apoptosis in cardiac cells [36]. Combined treatment with pembrolizumab and trastuzumab reduced cell viability and increased inflammatory markers IL-8 and IL-1β, but not IL-6, showing the complexity of cardiotoxic TKI responses.

RNA sequencing from rat cardiomyocytes treated with the TKI, Sotorasib, indicate pro-inflammatory responses, indicated by IL6/STAT3 and IL2/STAT5 signaling, as well as DNA damage response (UV Response) (Figure 4A, data adapted from GSE222642). Encode analysis also indicated an IRF8 signature, suggesting presence of inflammation, but also a SOX2 signature, demonstrating an oxidative stress response, and SMAD4, indicating TGF- β response (Figure 4B). These findings highlight the complex cardiotoxic effects of kinase inhibitors like sorafenib, which include oxidative stress, inflammation, and DNA damage responses in cardiac cells.

#### 2.1.4. Transcriptomics Analysis of the Indisulam Cardiotoxicity Signatures

Indisulam, also known as E7070, is a chloroindolyl sulfonamide that acts as a cell cycle inhibitor with antitumor properties. It has shown effectiveness in inhibiting the progression of various human tumor cells, including those in melanomas and blood-borne cancers like leukemia. Indisulam works by targeting the G1 phase of the cell cycle, depleting cyclin E, inducing p53 and p21, and inhibiting CDK2, which blocks the transition from the G1 to the S phase. Common side effects of indisulam include fatigue, nausea, and low blood cell counts. Cardiac side effects, such as arrhythmias and heart failure, have also been reported [37]. RNA sequencing from human cardiomyocytes treated with Indisulam, indicate pro-inflammatory responses, revealed by a IL17A signature. Statin and cholesterol pathway alterations were also observed (Figure 5A), indicating alterations of the lipid metabolism. Encode analysis also indicated a STAT3 signature, indicating inflammation, but also SOX2 signature, indicating oxidative stress response, as well as Calcium and metabolic alterations (Figure 5B).

#### 2.1.5. Transcriptomics Analysis of the Trastuzumab Cardiotoxicity Signatures

Trastuzumab is a monoclonal antibody therapy for HER2+ breast cancer, an aggressive type that makes up about 25% of cases. While it improves outcomes, its cardiotoxicity is a concern, with 10–15% of patients developing cardiomyopathy and 2–4% experiencing heart failure. This leads to 20–30% of patients discontinuing treatment, risking inadequate cancer therapy and recurrence. RNA sequencing from human cardiomyocytes treated with Trastuzumab, indicate pro-inflammatory responses, indicated by TNF-α signaling, as well as DNA damage response (5GY radiation) (Figure 6A). Encode analysis also indicated a ROS1 signature, indicating dysregulation of receptor tyrosine kinase signaling and LRRK2, indicating alteration of other kinase cascades (Figure 6B). In summary, while trastuzumab is effective in treating HER2+ breast cancer, its potential for causing heart-related side effects requires careful monitoring and management to ensure patients receive the full benefit of the therapy without compromising their health.

#### 2.1.6. Multi-OMICs Analysis and Targeting of the 5-Fluorouracil (5-FU) Cardiotoxicity

5-Fluorouracil (5-FU) is a chemotherapy drug used to treat various cancers, including those of the head, neck, esophagus, stomach, and colon. Despite its effectiveness, 5-FU can cause cardiotoxicity, which is a significant concern. The incidence of 5-FU-induced cardiotoxicity ranges from 1% to 19%, with angina pectoris being the most common symptom due to coronary vasospasm. Other serious cardiac events include myocardial infarction, arrhythmias, heart failure, acute pulmonary edema, pericarditis, and sudden cardiac death [38]. The mechanisms behind 5-FU cardiotoxicity are not fully understood but include coronary vasospasm, direct myocardial toxicity, oxidative stress, and thrombosis. Coronary vasospasm leads to reduced blood flow and ischemia, while direct myocardial toxicity involves damage to myocardial and endothelial cells. Oxidative stress results from increased ROS production, and thrombosis is due to endothelial damage. Management of 5-FU cardiotoxicity involves routine cardiac monitoring, dose adjustment, and supportive therapies like nitrates or calcium channel blockers to manage vasospasm. Recognizing and addressing 5-FU cardiotoxicity is crucial to prevent permanent damage and ensure effective cancer treatment.

Transcriptomics from rat cardiomyocytes treated with 5-Fluorouracil, indicate alterations of the eicosanoid lipid metabolism, a process dependent on Cyclooxygenase (COX) Pathway, as well as changes in mRNA processing (Figure 7B). TRRUST analysis also indicated a PPARA and PPARD signature, transcription factors involved in the regulation of lipid metabolism, but also TBX3 and TWIST2 signatures, factors contributing to inflammation (Figure 7A). In conclusion, while 5-FU is an effective chemotherapy agent, its potential cardiotoxic effects necessitate careful monitoring and management. Understanding the molecular mechanisms underlying these effects, such as alterations in lipid metabolism and inflammatory pathways, can help develop strategies to mitigate these risks and improve patient outcomes.

Furthermore, a recent study investigated how resveratrol can protect against 5-FU-induced cardiotoxicity. The researchers found that resveratrol reduces heart damage by inhibiting a type of cell death known as ferroptosis, which is dependent on the enzyme GPX4. This protective effect was observed both in a mouse model and in cultured cardiac cells, suggesting that resveratrol could be a potential treatment to mitigate the harmful side effects of 5-FU on the heart [39]. Another study confirmed the cardioprotective effect of resveratrol in treating gastric cancer. Liu et al. found that co-administration of resveratrol and 5-FU effectively reduced the viability, migration, and invasion of gastric cancer cells, while also decreasing tumor weight and volume [16]. Mechanistically, the combination induced p53-mediated apoptosis and autophagy, enhancing the anti-tumor effect. Finally, resveratrol mitigated the cardiotoxic effects of 5-FU by reducing oxidative stress and apoptosis in cardiomyocytes, suggesting its potential as an adjunct therapy to improve cancer treatment outcomes and protect against 5-FU-induced cardiotoxicity.

#### 2.1.7. Targeting of the Paclixatel-Mediated Cardiotoxicity

Paclitaxel is a chemotherapy drug used to treat various cancers, including breast, ovarian, and lung cancer. Despite its effectiveness, paclitaxel can cause cardiotoxicity, which is a significant concern. The incidence of paclitaxel-induced cardiotoxicity is relatively low but includes serious cardiovascular events. These events can range from bradycardia and atrial and ventricular arrhythmias to more severe conditions like congestive heart failure and myocardial infarction [40]. The mechanisms behind paclitaxel cardiotoxicity are not fully understood but include indirect effects such as massive histamine release leading to conduction disturbances and arrhythmias. Direct myocardial damage is also a factor, potentially affecting subcellular organelles and inducing congestive heart failure [41]. Additionally, paclitaxel can promote the formation of doxorubicinol metabolites when used with doxorubicin, contributing to cardiotoxicity. Management of paclitaxel cardiotoxicity involves careful monitoring of cardiac function, dose adjustment, and supportive therapies to mitigate adverse effects. Recognizing and addressing paclitaxel cardiotoxicity is crucial to prevent permanent damage and ensure effective cancer treatment. While no OMICs studies could be found yet concerning Paclitaxel-mediated cardiotoxicity, a recent study demonstrated that liposomal paclitaxel significantly reduces hematopoietic complications, such as neutropenia and lymphocytopenia, compared to Taxol [42]. Additionally, the liposomal formulation results in fewer cardiovascular side effects, including a lower incidence of cardiotoxicity and arrhythmias. The study also noted that liposomal paclitaxel maintains therapeutic efficacy while offering improved stability, making it a safer alternative for patients requiring paclitaxel-based chemotherapy.

### 2.2. Post-Translational Modifications (PTMs) Associated with Cardiotoxicity

While these transcriptomic approaches provided valuable insights into the molecular mechanisms underlying the adverse cardiac effects of various chemotherapeutic agents, more insights into these mechanisms through advanced techniques at the protein level are needed. Indeed, many of the targets (e.g., inflammatory cytokines) and pathways (e.g., cell senescence, cell transformation) identified are regulated by posttranslational modifications, such as ubiquitination and phosphorylation. Such specific protein modifications, if restricted to diseased hearts could be targeted for cardiac-specific therapy. Regulators of protein modification, such as ubiquitination or phosphorylation, are also essential for normal cardiac function [43,44] as well as regulation of the aforementioned signaling pathways [44,45,46], indicating a potential role in pathogenesis. Protein modifications play a crucial role in the development of cardiotoxicity, particularly in the context of cardiovascular diseases and treatments. Post-translational modifications (PTMs) such as phosphorylation, ubiquitination, and glycosylation can significantly alter protein function, stability, and interactions. These modifications are often implicated in the pathophysiology of cardiotoxicity by affecting key proteins involved in cardiac function and stress responses. For instance, the dysregulation of PTMs can lead to mitochondrial dysfunction, oxidative stress, and impaired calcium homeostasis, all of which contribute to cardiac cell damage and heart failure. Understanding these modifications is essential for developing targeted therapies to mitigate cardiotoxic effects and improve cardiovascular health outcomes.

#### 2.2.1. Kinases Associated with Cardiotoxicity

Kinases play a critical role in many cellular processes by catalyzing the transfer of a phosphate group, usually from ATP to specific target molecules. This process, known as phosphorylation, is a key regulatory mechanism that affects the activity, localization, stability, and interactions of proteins. Recent studies revealed several kinases that could be relevant in the cardio-oncology field, especially considering the druggability of enzymes (Table 3). These kinases are well known to be activated or mutated in cardiac diseases. According to the literature, inhibiting PKCδ for which drugs (e.g., KAI-9803) were tested in clinical trials for reperfusion injury, GRK4 for which drugs (e.g., Atelonol, Metoprolol) were tested in clinical trial for hypertension and hypertrophic cardiomyopathy or ROCK1 for which a drug (e.g., Fasudil) is approved for hypertension in Asia could improve the heart condition after chemotherapy. This possibility was recently investigated by a recent article that discusses how Fasudil [47]. The study involved both in vivo (mice) and in vitro (H9C2 cells) experiments. Fasudil [47]. The protective effects of Fasudil are attributed to its antioxidant, anti-senescence, and anti-apoptotic properties. The fact that all the drugs listed below, are clinically tested indicate their proper bioavailability in patients, and that they could be repurposed for cardioprotection.

In line with these findings, Phosphoinositide 3-Kinase Gamma (PI3Kγ) inhibition appeared to protect against anthracycline-induced cardiotoxicity and reduce tumor growth [48]. The study found that inhibiting PI3Kγ in mice preserved cardiac function and protected against doxorubicin-induced cardiotoxicity. This protective effect was linked to enhanced autophagic disposal of damaged mitochondria (i.e., mitophagy). Additionally, PI3Kγ inhibition synergized with doxorubicin’s anticancer effects by boosting anticancer immunity. Thus, PI3Kγ inhibition offers a dual therapeutic advantage by preventing cardiotoxicity and enhancing the anticancer efficacy of anthracyclines.

Finally, a recent abstract described the role of the PCSK9 inhibitor evolocumab in reducing cardiotoxicity caused by the sequential treatment of doxorubicin and trastuzumab. The study found that evolocumab exerts cardioprotective effects by enhancing cell viability and reducing cardiotoxicity through the MyD88/NF-κB/mTORC1 pathways [49]. This protective effect is linked to the inhibition of pro-inflammatory pathways and the enhancement of mitophagy. The findings suggest that evolocumab could be a promising therapeutic strategy to mitigate the cardiotoxic effects of anthracyclines and HER2-blocking agents in cancer patients.

#### 2.2.2. Ubiquitin Ligases Associated with Cardiotoxicity

Ubiquitination plays an important role in regulating protein degradation and cellular stress responses. Recent studies have described several ubiquitin ligases whose (dys)function is associated with cardiovascular disease, and for which drugs are available (Table 4). This list includes mostly enzymes from the NEDD4 family of ubiquitin ligases such as WWP2, that cause heart fibrosis with aging. WWP2 is a regulator of SMAD2 levels, an effector of the TGF-β pathway, that also regulates cardiac fibrosis in patients, for which drugs are available. WWP2 also regulates RAP1 function in the cardiac endothelium [50].

Several additional ubiquitin ligases could be targeted for cardioprotection, such as LZTR1 that is mutated in the cardiac syndrome called Noonan Syndrome [54], as well as the β-catenin signaling regulator, PDZRN3 which is also important for cardiovascular development in mice [55,56]. TRAF6 also appears to be a major regulator of cardiac inflammation by modulating NF-κB signaling [53].

In line with these findings, a recent article investigated the role of TRIM25 in reducing doxorubicin-induced cardiotoxicity [57]. The study found that the ubiquitin ligase, TRIM25 helps mitigate this cardiotoxicity by degrading p85α, a regulatory subunit of PI3K. This degradation leads to increased nuclear translocation of XBP-1s, which reduces endoplasmic reticulum stress and apoptosis in cardiomyocytes. The findings suggest that TRIM25 could be a potential therapeutic target for protecting the heart during doxorubicin treatment and further strengthens the evidence fora role of PI3K signaling in cardioprotection.

#### 2.2.3. Ubiquitin Hydrolases Associated with Cardiotoxicity

OTUB1, a deubiquitinating enzyme with multiple substrates [58], acts as a critical player in variouscardiovascular diseases [59], including doxorubicin-induced cardiotoxicity [60]. This cardiotoxicity primarily arises from increased oxidative stress and apoptosis in cardiomyocytes. By stabilizing key proteins involved in antioxidant defense and cell survival, OTUB1 effectively reduces oxidative damage and prevents cardiomyocyte apoptosis [61]. Moreover, transcriptomics analysis comparing *Otub1* WT and KO mouse cardiomyocytes after anthracycline treatment [60] revealed changes in the STAT1 pathway, indicating differential inflammatory responses (Figure 8A), as well as a clear differential interferon response (Figure 8B). The protective mechanism of OTUB1 underscores its potential as a therapeutic target to alleviate the cardiotoxic side effects of doxorubicin [60].

In line with these findings, a recent article investigated the role of Ubiquitin-Specific Protease 36 (USP36) in doxorubicin-induced cardiomyopathy [62]. The study found that USP36 expression increases in cardiomyocytes exposed to doxorubicin, leading to oxidative stress and apoptosis. Silencing USP36 significantly mitigated these harmful effects both in vitro and in vivo. Mechanistically, USP36 interacts with and stabilises Poly (ADP-ribose) polymerase 1 (PARP1), preventing its ubiquitination. This interaction increases PARP1 levels, contributing to the cardiotoxic effects of doxorubicin. In a mouse model, cardiac knockdown of USP36 preserved cardiac function and protected against structural changes in the heart caused by doxorubicin. The findings suggest that the USP36/PARP1 axis plays a significant role in the pathogenesis of DIC and could be a potential therapeutic target for mitigating doxorubicin-induced cardiotoxicity.

### 2.3. Identification of Therapeutic Targets Using Multi-OMICs for Cardioprotection

We have listed in the table below, known kinases and ubiquitin ligases known to regulate different molecular functions important for cardiac function (Table 5). Such information is key to developing well-tolerated and appropriate therapeutic targeting approaches for cardio protection. Understanding the regulation of these pathways by ubiquitin ligases and kinases can provide valuable insights into the mechanisms of cardiotoxicity and may lead to the development of novel therapeutic strategies for the treatment of heart disease.

#### 2.3.1. Targeting the TGF-β Pathway for Cardioprotection

TGF-β is a multifunctional cytokine involved in various cellular processes, including fibrosis and inflammation. Chemotherapeutic drugs often promote the overexpression of TGF-β, leading to myocardial fibrosis, which impairs heart function and increases the risk of heart failure. Recent studies have shown that TGF-β inhibitors, such as small molecule inhibitors and monoclonal antibodies that specifically target TGF-β or its receptors [63]. This approach not only targets the cancer cells but also protects the heart, offering a dual benefit in oncology care [64] Another approach focuses on blocking the downstream signaling pathways activated by TGF-β, such as the Smad proteins, which play a crucial role in mediating its fibrotic effects. Overall, the development of TGF-β inhibitors and Smad protein blockers represents a promising area of research in the field of cardio-oncology. By targeting the molecular mechanisms underlying myocardial fibrosis, these agents may help to reduce the risk of heart failure and improve outcomes for cancer patients.

#### 2.3.2. Targeting the RELA/NF-κB Pathway for Cardioprotection

RELA, also known as p65, is a subunit of the NF-κB complex that plays a crucial role in regulating inflammatory responses and cell survival. Activation of the NF-κB/RELA pathway can lead to the expression of genes involved in inflammation, apoptosis, and fibrosis [65], which are critical factors in the development of cardiotoxicity. By inhibiting the RELA pathway, it is possible to reduce the inflammatory and fibrotic responses in the heart, thereby protecting cardiomyocytes from damage. This approach has shown potential in preclinical studies, where pharmacological inhibitors of RELA have been able to mitigate cardiac injury and improve heart function. Additionally, targeting RELA can help in modulating the adverse effects of chemotherapeutic agents, such as anthracyclines, which are known to induce cardiotoxicity through oxidative stress and inflammatory pathways [66].

Furthermore, a recent article described the discovery of a small-molecule inhibitor, C25-140, that targets the TRAF6-Ubc13 interaction (Table 3), which is crucial for NF-κB signaling involved in inflammation and autoimmunity [53]. The inhibitor was found to reduce TRAF6 activity, thereby decreasing NF-κB activation in various immune and inflammatory pathways. This compound showed promising results in preclinical models of autoimmune diseases like psoriasis and rheumatoid arthritis, where it significantly reduced inflammation and improved disease outcomes. Given its ability to inhibit NF-κB signaling, C25-140 could potentially be useful for treating cardiac inflammation. NF-κB is a key player in the inflammatory response, and its inhibition could help reduce inflammation in cardiac tissues, thereby protecting against further damage and improving heart function. This could be particularly beneficial in conditions like myocarditis, where inflammation plays a significant role in disease progression.

#### 2.3.3. Targeting the ADAM15/TNF Pathway for Cardioprotection

ADAM17, also known as tumor necrosis factor α converting enzyme (TACE), is crucial for the activation of pro-inflammatory TNF-α through its cleavage. This suggests that inhibiting ADAM17 could potentially benefit conditions where TNF-α plays a key role. However, the link between ADAM17 and doxorubicin-induced cardiomyopathy is still unclear [67]. RNA sequencing analysis of cardiomyocytes from ADAM17 knockout mice vs. controls showed significant changes in inflammatory signaling pathway genes, such as NFATC1 and NFATC2 or Type III interferon signaling, indicating that knockdown of this gene could reduce cardiac inflammation (Figure 9A,B). Further research has demonstrated that ADAM17 plays a role in cardiomyocyte apoptosis and cardiac remodeling induced by doxorubicin [68]. Inhibiting ADAM17 can alleviate these adverse effects by reducing the activation of pro-inflammatory pathways and decreasing oxidative stress. This protective mechanism highlights the potential of ADAM17 as a therapeutic target for mitigating doxorubicin-induced cardiotoxicity.

#### 2.3.4. Targeting the EPAC1/RAP1 Pathway for Cardioprotection

Targeting EPAC1 (Exchange Protein directly Activated by cAMP 1) has emerged as a promising strategy for cardioprotection [69]. EPAC1 plays a crucial role in mediating the effects of cyclic adenosine monophosphate (cAMP) within cardiac cells, influencing various cellular processes such as calcium handling, cell survival, and inflammation. Research has shown that EPAC1 activation can mitigate cardiac fibrosis and hypertrophy whereas pharmacological inhibition of EPAC1 has been demonstrated to protect against cardiac dysfunction induced by pressure overload and ischemia-reperfusion injury. A recent study on EPAC1 inhibition highlights its potential as a therapeutic strategy for cardioprotection upon doxorubicin [70]. Inhibition ofEPAC1, either pharmacologically or genetically, can significantly mitigate the adverse effects of doxorubicin on the heart. EPAC1 inhibition not only prevents DNA damage and mitochondrial dysfunction in cardiomyocytes but also enhances the anticancer efficacy of doxorubicin. Furthermore, EPAC1 directly regulates RAP1 function. Rap1, a small GTPase, is involved the progression of atherosclerosis and cardiac diseases [50]. These findings suggest that modulating the EPAC1/RAP1 signaling activity could offer a novel therapeutic approach to prevent and treat cardiovascular diseases, enhancing the resilience of the heart to various forms of stress.

#### 2.3.5. Targeting the AMPK Pathway for Cardioprotection

The activation of NRF2 and AMPK has been shown to have protective effects on cardiomyocytes, which are crucial for maintaining heart health. NRF2 is a key regulator of the antioxidant response, and its activation has been shown to reduce ROS (reactive oxygen species) levels and inhibit the mitochondrial apoptotic pathway, thereby protecting cardiomyocytes from damage. AMPK (AMP-activated protein kinase) activation has also been shown to have beneficial effects on the heart. It enhances mitochondrial biogenesis, promotes oxidative metabolism, and reduces apoptosis (cell death) and fibrosis (scarring), all of which can contribute to improved cardiac function and survival [71]. Studies have demonstrated that activating AMPK through various agents, such as metformin and resveratrol, can provide significant cardioprotective effects [72]. These findings are important for developing targeted therapies that address both the cardiotoxicity (heart damage) and anti-cancer efficacy of anthracyclines, a class of chemotherapy medications. Anthracyclines are effective against certain types of cancer, but they can also cause damage to the heart. By activating NRF2 and AMPK, it may be possible to reduce the cardiotoxic effects of anthracyclines while preserving their anti-cancer effects.

#### 2.3.6. Additional Pathways that Could Be Targeted for Cardioprotection

The actin cytoskeleton pathway is a dynamic structure that plays a crucial role in maintaining the shape and function of cardiac cells. The ubiquitin ligase TITIN regulates this pathway, and dysregulation can lead to decreased contractility of the heart [73]. The calcium signaling pathway is critical for regulating the electrophysiology and contractility of the heart. The ubiquitin ligase PKD1 and the kinases CAMK2 and GRK5 regulate this pathway. Dysregulation of this pathway can lead to arrhythmias and decreased contractility of the heart. The oxidative stress pathway is involved in the regulation of reactive oxygen species (ROS) production, which can lead to cellular damage and apoptosis. The ubiquitin ligase ITCH, through its regulation of TXNIP, and the kinase ASK1 regulate this pathway. The AKT/ERK signaling pathway is involved in the regulation of cardiac hypertrophy, a condition characterized by the thickening of the heart muscle [74]. The ubiquitin ligases NEDD4-1 and LZTR1, and the kinases MEK-1/2 and ERK-5 regulate this pathway.

The PTEN signaling pathway is involved in the regulation of cell survival and apoptosis. The ubiquitin ligases NEDD4-1 and WWP1, and the kinases GRK-4, GSK-3, B-RAF, and ROCK regulate this pathway. Dysregulation of this pathway can lead to decreased contractility and apoptosis of cardiac cells. The cell cycle pathway is involved in the regulation of cell proliferation and is critical for maintaining the homeostasis of cardiac cells. The ubiquitin ligases NEDD4-L, SMURF1, and SMURF2, and the kinase CDK-9 regulate this pathway [52]. Dysregulation of this pathway can lead to cardiac hypertrophy. The apoptosis pathway is involved in the regulation of programmed cell death, which is critical for maintaining the homeostasis of cardiac cells. The ubiquitin ligases WWP2 and MKRN1, and the kinase CK1α regulate this pathway. Dysregulation of this pathway can lead to cardiac necrosis. The inflammation pathway is involved in the regulation of the inflammatory response, which is critical for maintaining the homeostasis of cardiac cells. The ubiquitin ligase TRAF-6 and the kinases IκB and PCSK9 regulate this pathway [75]. Dysregulation of this pathway can lead to cardiac inflammation.

## 3. Discussion

The improved survival rates of cancer patients due to advanced therapies have paradoxically led to a growing concern in cardio-oncology, as these treatments are also associated with an increased risk of long-term cardiovascular complications [1]. Hence, the field aims to develop strategies to prevent and treat these cardiovascular complications. This includes the use of cardioprotective agents, lifestyle modifications, and regular cardiovascular monitoring for cancer survivors. By integrating cardiology and oncology care, clinicians aim to improve the overall health outcomes and quality of life for cancer survivors [4]. This interdisciplinary approach is crucial as it addresses the complex interplay between cancer treatments and cardiovascular health, ensuring that patients receive comprehensive care throughout their cancer journey and beyond. To advance our scientific insights into the molecular mechanisms of chemotherapy-induced cardiotoxity, more advanced techniques at the protein level are needed to study protein activity and PTMs such as phosphorylations and ubiquitinations. Indeed, multi-omics approaches, such as combining transcriptomics, proteomics and metabolomics, will further facilitate the identification of new therapeutic targets for cardioprotection upon chemotherapy.

The analysis of cardiotoxicity using comprehensive omics approaches, such as genomics, transcriptomics, proteomics, and metabolomics, has provided valuable insights into the molecular mechanisms underlying the adverse cardiac effects of various chemotherapeutic agents. The text highlights the power of these unbiased, multi-omics analyses in elucidating the complex pathways and signaling cascades that are dysregulated in cardiomyocytes and cardiac endothelial cells upon exposure to chemotherapy drugs. The key findings from these OMICs studies include the activation of pro-inflammatory pathways, disruption of critical signaling networks (such as NFKb, SP1, SP3, JAK-STAT pathways), alterations in DNA damage repair and chromatin remodeling genes, and changes in lipid metabolism and oxidative stress responses. These molecular insights are crucial for understanding the multifaceted nature of chemotherapy-induced cardiotoxicity and developing targeted strategies to mitigate these adverse effects.

## 4. Conclusions

Moving forward, the integration of these omics data with clinical observations and patient-specific models, such as induced pluripotent stem cell-derived cardiomyocytes, will further enhance our understanding of individual variability in drug responses and susceptibility to cardiotoxicity. This knowledge can guide the design of personalized treatment approaches, where the anti-cancer efficacy of these drugs is maintained while their cardiotoxic impact is minimized. Ultimately, the comprehensive analysis of cardiotoxicity using multi-omics techniques represents a powerful tool for elucidating the complex mechanisms underlying drug-induced cardiac dysfunction. By leveraging these insights, researchers and clinicians can work towards developing more effective and safer cancer therapies that improve patient outcomes and quality of life. In conclusion, here we provide a comprehensive overview of the cardiotoxic effects associated with various chemotherapy drugs, highlighting their mechanisms, relevant OMICs findings, and potential interventions (Table 6). This is a first step towards the integration of the different studies carried out on the molecular profiling cardiotoxicity of anti-cancer agents. This detailed analysis underscores the importance of tailored interventions to mitigate the cardiovascular risks associated with chemotherapy, ultimately enhancing patient outcomes and quality of life.

## Figures and Tables

**Figure 1 jox-15-00009-f001:**
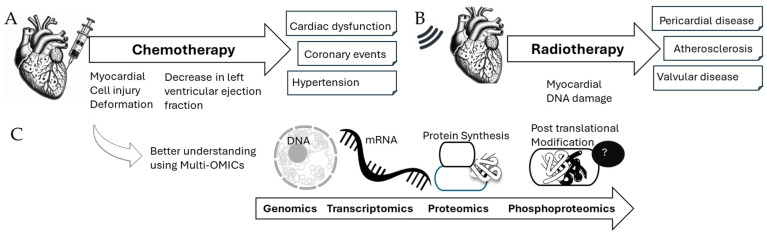
Cardiovascular events in cancer patients after chemotherapy or radiotherapy (**A**,**B**) (adapted from [17]) as well as relevance of multi-OMICs analyses to improve their understanding (**C**).

**Figure 2 jox-15-00009-f002:**
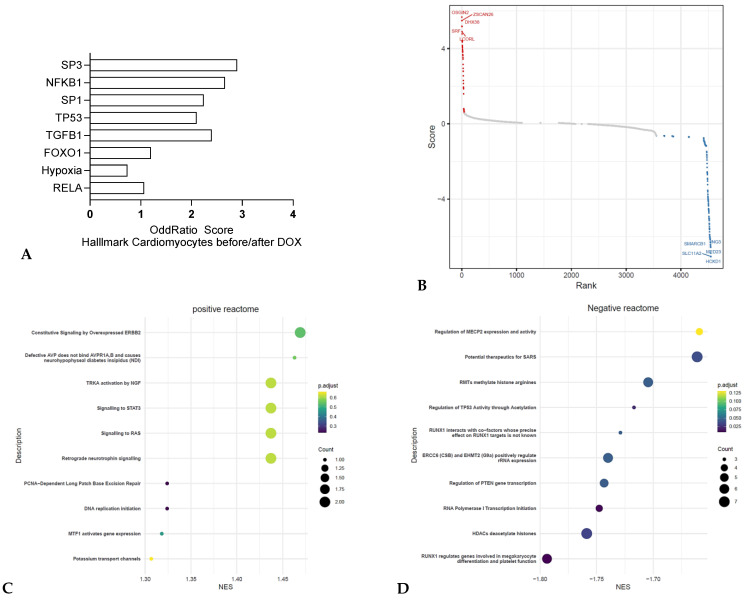
TrancriptOMICs analysis of cardiac disease initiation in cardiomyocytes (**A**) Pathway analysis of RNA sequencing from isolated cardiomyocytes after exposition to doxorubicin (GSE226116) comparing differentially expressed genes (q value < 0.005) using ENRICHR (Hallmark analysis). N = 3. Odds ratios are graphed, and the bars sorted from the most significant adjusted *p* value. (**B**) CRISPR screen comparing inhibited sgRNA (CRISPRi) when comparing the Doxocyclin-treated vs. Vehicle IPSC-derived cardiomyocytes conditions (Dataset: GSE276161). Positively and negatively enriched sgRNAs in red and blue respectively, grey indicate sgRNAs that do not show effects. (**C**,**D**) Reactome analysis performed on the positively and negatively enriched sgRNAs. The color of the dot corresponds to the adjusted *p*-value for each pathway (green for the most significant adjusted *p* value to dark blue for the less significant ones) and the size of the dot corresponds to the number of inhibited sgRNA for each group.

**Figure 3 jox-15-00009-f003:**
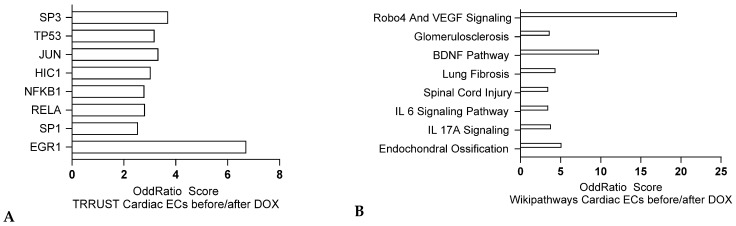
Transcriptomic analysis of cardiac disease initiation in endothelial cells. (**A**,**B**) Pathway analysis of RNA sequencing from isolated cardiac ECs after exposition to doxorubicin (GSE226116) comparing differentially expressed genes (q value < 0.005) using ENRICHR N = 3. Odds ratios are graphed, and the bars are sorted from the most significant corrected *p* value. TRRUST analysis and Wikipathways analysis respectively.

**Figure 4 jox-15-00009-f004:**
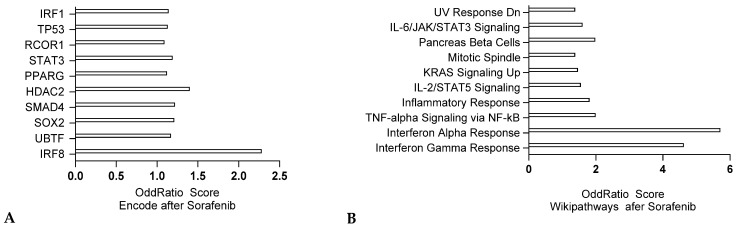
Transcriptomic analysis of cardiac disease initiation upon Sorafenib treatment in cardiomyocytes adapted from Series GSE222642 comparing differentially expressed genes (q value < 0.005), where male rats were gavaged with 50 mg/kg sorafenib (heart tissues collected at 14 days after treatment). ENCODE analysis (**A**) and Wikipathways analysis (**B**) respectively (ENRICHR, software developed by Ma’ayan lab, Computational Systems Biology, New York, NY, USA).

**Figure 5 jox-15-00009-f005:**
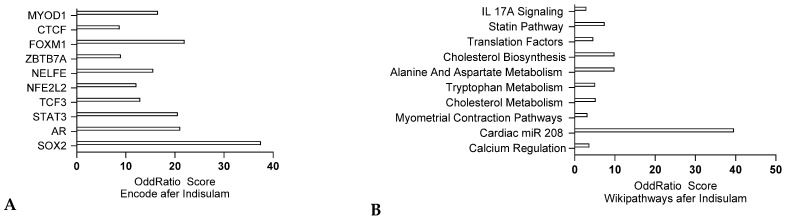
Transcriptomic analysis of cardiac disease initiation upon Indisulam treatment in cardiomyocytes from Query DataSets for GSE213311 comparing differentially expressed genes (q value < 0.005), RNA-seq analysis on cardiomyocytes treated with vehicle or indisulam. ENCODE analysis (**A**) and Wikipathways analysis (**B**) respectively (ENRICHR, Ma’ayan lab).

**Figure 6 jox-15-00009-f006:**
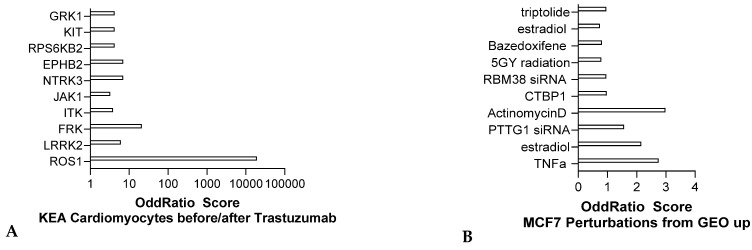
Transcriptomic analysis of cardiac disease initiation upon Trastuzumab treatment in cardiomyocytes adapted from GSE264120 comparing differentially expressed genes (q value < 0.005), ENCODE analysis (**A**) and MCF7 GEO UP signatures analysis (**B**) respectively (ENRICHR, Ma’ayan lab).

**Figure 7 jox-15-00009-f007:**
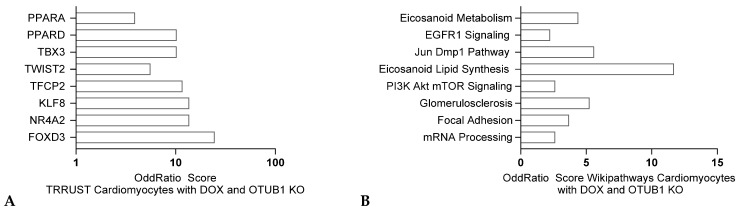
Transcriptomic analysis of cardiac disease initiation in rats upon 5-FU treatment in cardiomyocytes adapted from GSE166957 comparing differentially expressed genes (q value < 0.005), TRRUST analysis (**A**) and Wikipathways analysis (**B**) respectively (ENRICHR, Ma’ayan lab).

**Figure 8 jox-15-00009-f008:**
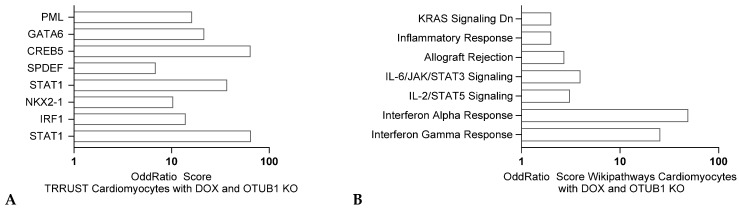
Transcriptomic analysis of cardiomyocyte response to doxorubicin in mice with OTUB1 heterozygous knockout according to Data obtained from GSE240959 and comparing differentially expressed genes (q value < 0.005), using ENRICHR. Odds ratios are graphed, and corrected *p* values are indicated. TRRUST analysis (**A**) and Wikipathways analysis (**B**) respectively.

**Figure 9 jox-15-00009-f009:**
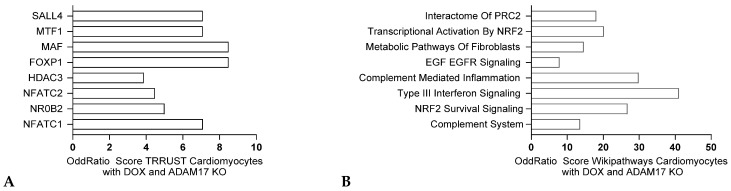
Transcriptomic analysis of cardiomyocyte response to doxorubicin in mice with ADAM17 knockout. Data obtained from GSE276325 comparing differentially expressed genes (q value < 0.005), Odds ratios are graphed, and bars are sorted from the most significant corrected *p*-value. TRRUST analysis (**A**) and Wikipathways analysis (**B**) respectively.

**Table 2 jox-15-00009-t002:** Cardiovascular impact of common anti-cancer drugs.

Anticancer Drug	Mechanism of Action	Cardiotoxic Effects
**Anthracyclines**	Disrupt DNA replication and increase ROS generation	Mitochondrial dysfunction, cardiomyocyte apoptosis
**Antimetabolites (e.g., 5-FU)**	Induce oxidative stress and inflammation, alter biochemical pathways	Myocytic degeneration, endothelial injury
**Alkylating Agents (e.g., Cyclophosphamide, Cisplatin)**	Modify DNA, RNA, and proteins, leading to oxidative stress and inflammation	Severe oxidative stress, inflammation, cardiotoxicity
**Microtubule Inhibitors (e.g., Paclitaxel)**	Inhibit mitosis by altering microtubule polymerization	Endothelial damage, cardiotoxicity
**Tyrosine Kinase Inhibitors (e.g., Imatinib, Sorafenib)**	Disrupt signal transduction and angiogenesis	Cardiotoxic effects, particularly in HER2-positive breast cancer treatments
**Proteasome inhibitors**	Affect protein degradation by blocking the proteasome	Causes protein aggregation, endoplasmic reticulum stress, and autophagy

**Table 3 jox-15-00009-t003:** Kinases associated to cardiovascular phenotypes.

Kinases	Disease/Function	Inhibitor	Clinical Trial ID
**ALK1**	Hemorrhages, hypertension AVMs	Lorlatinib	NCT03127618
**BMPR2**	Pulmonary hypertension	Fostamatinib	NCT01608542
**CAMK2**	Cardiac arrhythmias	Bosutinib	NCT02192294
**CDK9**	Cardiac hypertrophy	Flavopiridol	NCT00045448
**GRK4**	Hypertension	Atenolol, Metoprolol	NCT01736488, NCT04133532
**GSK3**	Cardiac hypertrophy	NP031112	NCT03692312
**PKCα**	Heart contractility	LY-900003, Safingol, Go6976.	NCT02826759
**PKCδ**	Ischemic-reperfusion injury	KAI-9803	NCT00785954
**PKCε**	Cardiac ischemia	KAI-1678	NCT01135108
**ROCK**	Hypertension	Fasudil	Approved Japan/China
**TITIN**	Cardiomyopathies	Flavin Mononucleotide	NCT04179604
**WNK1-4**	Hypertension	Hydrochlorothiazide	NCT03946514

**Table 4 jox-15-00009-t004:** Ubiquitin ligases associated to cardiovascular phenotypes.

Ubiquitin Ligases	Disease/Function	Inhibitor
**NEDD4**	Heart development	Described in [51]
**NEDD4L**	Cardiac voltage-gated Na+ channels function	
**SMURF1/2**	Cardiac remodeling/Fibrosis	
**ITCH**	Cardiac inflammation	Described in [45]
**WWP1/WWP2**	Perivascular Fibrosis/endothelial damage [50]	I3C [52]
**LZTR1**	Noonan Syndrome	
**PDZRN3**	Cardiac hypertrophy	
**MIB1**	Left ventricular noncompaction	
**MID1**	Ventral/atrial septal defects/BBB syndrome	
**TRIM13**	ER stress autophagy upon cardiac failure	
**TRAF6**	Cardiac inflammation	Described in [53]
**bTRCP/FBXW1**	Anti-angiogenic	

**Table 5 jox-15-00009-t005:** Kinases and Ubiquitin ligases associated to pathways, important for cardioprotection.

Cardiotoxicity Altered Pathways	Function in Heart Disease	Known Regulators—Ubiquitin Ligases	Known Regulators—Kinases
**TGF-β** **signaling**	Cardiac fibrosis	WWP2, WWP1, ITCH	WNK-1/5, BMPR2, ALK
**Junctional defects/** **β-catenin signaling**	Decrease contractility	PDZRN3	PKC-α/δ/ε
**Actin cytoskeleton**	Decrease contractility		TITIN
**Calcium signaling**	Electrophysiology/Contractility	PKD1	CAMK2, GRK5
**Oxidative stress**	Arrhythmia	ITCH (through TXNIP)	ASK1
**AKT/ERK signaling**	Cardiac hypertrophy	NEDD4-1, LZTR1	MEK-1/2, ERK-5
**PTEN signaling**	Decrease contractility/apoptosis	NEDD4-1, WWP1	GRK-4, GSK-3, B-RAF, ROCK
**Cell Cycle (Cyclins/CDK)**	Cardiac hypertrophy	NEDD4-L, SMURF1/2	CDK-9
**Apoptosis (PARP1)**	Cardiac necrosis	WWP2, MKRN1	CK1α
**Inflammation (NFκB)**	Cardiac inflammation	TRAF-6	IκB, PCSK9

**Table 6 jox-15-00009-t006:** Summary table.

Chemotherapy Drugs and Cardiotoxicity	Mechanisms	OMICs Findings	Potential Interventions
**Cytostatic Agents (Cisplatin, Taxanes, Anthracyclines)**	DNA damage, oxidative stress	Increased oxidative stress and inflammation markers	Antioxidants, cardioprotective agents, Antiinflammatory therapy
**Antimetabolites (5-FU)**	Induce oxidative stress and inflammation	Alterations in lipid metabolism, oxidative and inflammatory pathways	Resveratrol that blocks ferroptosis was tested
**Tyrosine Kinase Inhibitors (Sunitib, Sorafenib)**	Inhibition of kinase of the EGFR pathways	Increased DNA damage and inflammation markers	Cardiac monitoring, dose adjustment
**Trastuzumab**	HER2 blocking antibody	Induced pro-inflammatory responses, DNA damage response	Antiinflammatory therapy
**Paclixarel**	Inhibit mitosis by altering microtubule polymerization	Not defined	Liposomal formulation causes less cardiovascular side effects, when compard to taxol
**Immune Checkpoint Inhibitors (Pembrolizumab, Nivolumab)**	Immune-mediated myocarditis	Immune response gene activation, cytokine release profiles	Immunosuppressive therapy
**VEGF Inhibitors (Bevacizumab, Nintedanib)**	Vascular endothelial dysfunction	Altered angiogenesis-related gene expression, endothelial cell damage markers	Vascular protective agents
**Proteasome Inhibitors (Bortezomib, Carfilzomib)**	Disruption of protein homeostasis	Accumulation of ubiquitinated proteins, endoplasmic reticulum stress	Cardioprotective agents

## Data Availability

No new data were created or analyzed in this study.

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
