# Peer review of "Cardiotoxicity of Chemotherapy: A Multi-OMIC Perspective"

_jox, 2025, doi:10.3390/jox15010009_

Round 1

Reviewer 1 Report

Comments and Suggestions for Authors

The topic discussed by the authors is significant both medically and socially, given the increasing prevalence of cancer worldwide. Fortunately, recent advancements in therapy offer improved treatment options, albeit often accompanied by side effects. Therefore, the focus of this work is well-chosen and addresses a critical need in modern medicine. While the paper is engaging and informative, several areas require clarification and improvement.

Specific Comments and Recommendations:

Line 118 and Others (Methodology):

The authors did not include a separate methods section or incorporate the increasingly popular PRISMA guidelines into their review. While the multi-OMIC perspective seems to fulfill the necessary methodological requirements, it would be necessary to  enhance the paper’s structure to clearly delineate the methodology chapter. This would provide clarity and improve transparency.

Line 28-30:

In the statement, "While 30% of cancer patients die of cancer itself, around 10% die from cardiovascular complications," the authors need to clarify whether these percentages overlap or are mutually exclusive. Are they suggesting that 40 % (30% + additional 10%) of cancer patients die from cancer and cardiovascular complications or 30%, i.e., 20% from cancer plus 10% from cardiovascular complications? It needs clarification.

Line 50 (Table):

The authors did not mention proteasome inhibitors as potential risk factors for cardiovascular diseases in Table 1. This omission is particularly notable in the context of multiple myeloma patients with cardiac (light chain) amyloidosis, where proteasome inhibitor therapy could exacerbate cardiac insufficiency. The same comment applies to Table 2 (line 103).

Page 7:

The authors failed to discuss Nebivolol, the only beta-blocker identified to potentially prevent chemotherapy-related heart failure. Although this association demonstrated only borderline significance and was specific to anthracycline- or capecitabine-treated patients, it warrants mention in a review of this scope. Including this information, with proper reservations and critical analysis, would enrich the discussion.

(See: György Fogarassy, MD, Journal of Cardiovascular Medicine, 2021; 22(6): 459-468.)

Author Response

Comment 1: Line 118 and Others (Methodology):

The authors did not include a separate methods section or incorporate the increasingly popular PRISMA guidelines into their review. While the multi-OMIC perspective seems to fulfill the necessary methodological requirements, it would be necessary to  enhance the paper’s structure to clearly delineate the methodology chapter. This would provide clarity and improve transparency.

Reply 1: We have now clarified the methodology used for the analysis of existing data in this paper. As this is a review, we cannot have a separate methodology section (lines 128 -133). 

Comment 2: Line 28-30:

In the statement, "While 30% of cancer patients die of cancer itself, around 10% die from cardiovascular complications," the authors need to clarify whether these percentages overlap or are mutually exclusive. Are they suggesting that 40 % (30% + additional 10%) of cancer patients die from cancer and cardiovascular complications or 30%, i.e., 20% from cancer plus 10% from cardiovascular complications? It needs clarification.

Reply 2: We have now clarified this statement. 

Reply 3: Line 50 (Table):

The authors did not mention proteasome inhibitors as potential risk factors for cardiovascular diseases in Table 1. This omission is particularly notable in the context of multiple myeloma patients with cardiac (light chain) amyloidosis, where proteasome inhibitor therapy could exacerbate cardiac insufficiency. The same comment applies to Table 2 (line 103).

Comment 3: We appreciate this comment, and have now completed both tables accordingly.

Page 7:

The authors failed to discuss Nebivolol, the only beta-blocker identified to potentially prevent chemotherapy-related heart failure. Although this association demonstrated only borderline significance and was specific to anthracycline- or capecitabine-treated patients, it warrants mention in a review of this scope. Including this information, with proper reservations and critical analysis, would enrich the discussion.

(See: György Fogarassy, MD, Journal of Cardiovascular Medicine, 2021; 22(6): 459-468.)

Comment 4: We have now added this very relevant information to the review ( line 174).

Reviewer 2 Report

Comments and Suggestions for Authors

The article is very well written, easy to read and understand, and the topic is very interesting. However, I think it could be improved in several minor aspects:

1 - The introduction to the article is very good, it is easy to understand and the article is very well framed.

2 - The main text is more complex, but it presents very relevant and new information. I think it could be simplified by summarising this information in a table.

3 - Figure 2 is incomprehensible, of poor quality and it is not clear where the data presented comes from. Are they the author's own and have they never been published? If not, should he ask permission to use them?

4 - The same goes for the other figures 3-9, but they are of good quality.

5 - I think it's important to put a summary figure where you can summarise the article.

6 - The discussion is very short, and could be included in the conclusions. 

7 - I think there could be a chapter on the most relevant preventive or interventional primary health measures for these patients undergoing chemotherapy and radiotherapy.

Author Response

1 - The introduction to the article is very good, it is easy to understand and the article is very well framed.

reply: We thank the reviewers for the comments. 

2 - The main text is more complex, but it presents very relevant and new information. I think it could be simplified by summarising this information in a table.

3 - Figure 2 is incomprehensible, of poor quality and it is not clear where the data presented comes from. Are they the author's own and have they never been published? If not, should he ask permission to use them?

Reply: We have now redesigned Figure 2 and added a summary of the  findings

4 - The same goes for the other figures 3-9, but they are of good quality.

Reply: We improved the figure legends for these figures to improve their understanding

5 - I think it's important to put a summary figure where you can summarise the article.

Reply: We have now added a summary.

6 - The discussion is very short, and could be included in the conclusions. 

Reply: We have now added more text in the discussion.

7 - I think there could be a chapter on the most relevant preventive or interventional primary health measures for these patients undergoing chemotherapy and radiotherapy.

Round 2

Reviewer 1 Report

Comments and Suggestions for Authors

The authors answered  all my remarks, with no other comments